# SUPERVISED BATCH NORMALIZATION

## ABSTRACT

Batch Normalization (BN), a widely-used technique in neural networks, enhances generalization and expedites training by normalizing each mini-batch to the same mean and variance. However, its effectiveness diminishes when confronted with diverse data distributions. To address this challenge, we propose Supervised Batch Normalization (SBN), a pioneering approach. We expand normalization beyond traditional single mean and variance parameters, enabling the identification of data modes prior to training. This ensures effective normalization for samples sharing common features. We define contexts as modes, categorizing data with similar characteristics. These contexts are explicitly defined, such as domains in domain adaptation or modalities in multimodal systems, or implicitly defined through clustering algorithms based on data similarity. We illustrate the superiority of our approach over BN and other commonly employed normalization techniques through various experiments on both single and multi-task datasets. Integrating SBN with Vision Transformer results in a remarkable *15.13%* accuracy enhancement on CIFAR-100. Additionally, in domain adaptation scenarios, employing AdaMatch demonstrates an impressive *22.25%* accuracy improvement on MNIST and SVHN compared to BN.

## 1 INTRODUCTION

In the realm of deep learning, input normalization is essential for optimizing the training process of deep neural networks (DNNs) by addressing the variations in feature magnitudes. This method has been shown to accelerate convergence in neural networks with a single hidden layer, as highlighted by LeCun et al. LeCun et al. (2002). However, its efficacy diminishes in more complex architectures with multiple hidden layers. This decline is due to the progressive transformation of data through successive layers, which causes activations to diverge from the properties of the initially normalized inputs. To address this challenge, normalizing activations during training has become a critical approach. By ensuring that the statistical properties of activations remain consistent across all layers, this strategy facilitates stable and efficient training of deep neural networks. Consequently, this practice not only enhances the convergence rate but also significantly improves the overall performance of the model.

Batch Normalization (BN) Ioffe & Szegedy (2015), a popular activation normalization technique, stabilizes the optimization process by normalizing feature statistics within a batch. Despite its widespread success, Batch Normalization (BN) has notable drawbacks due to its reliance on mini-batch statistics. While the variability in batch statistics can enhance robustness and generalization, it also leads to issues when the mean and variance estimates are inaccurate. This is particularly problematic with heterogeneous data and small batch sizes, which can cause BN to fail in effectively normalizing activations. In such cases, BN struggles to normalize activations using a single mean and variance Wu & He (2018); Bilen & Vedaldi (2017); Deecke et al. (2018).

To overcome these limitations, we introduce Supervised Batch Normalization (SBN). SBN assigns samples in a mini-batch to different modes using predefined groups called contexts, then normalizes each sample based onin neural networks, enhances generalization and expedites train- the statistics of its corresponding context. **Instead of relying on random mini-batches, SBN utilizes contexts that group similar samples through domain knowledge or clustering algorithms**. The proposed method can be seamlessly integrated as layers in standard deep learning libraries. We evaluated SBN on various classification tasks and demonstrated that it consistently outperforms BN and other widely used normalization techniques.

## 2 RELATED WORK

### 2.1 NORMALIZATION METHODS

Batch normalization (BN) Ioffe & Szegedy (2015) is the most common normalization technique in cutting-edge classification architectures. Recently, new alternatives have emerged to broaden its applicability and enhance its generalizability. Batch Renormalization Ioffe (2017) is an extension of BN that addresses the issue of varying mini-batch statistics during training. Weight Normalization Salimans & Kingma (2016) reparameterizes the weight vectors in a neural network by separating their magnitude and direction. This technique simplifies the optimization process and often results in faster convergence during training. It introduces additional parameters to stabilize training by aligning the statistics of the current mini-batch with the moving averages of the training data. Layer Normalization Ba et al. (2016) is a technique that normalizes samples across the features for each individual example, rather than across the min-batch. This approach helps stabilize the hidden states in recurrent neural networks and improves training efficiency by eliminating the dependency on mini-batch size. Instance Normalization Ulyanov et al. (2016) normalizes samples across each feature map for individual examples, making it particularly effective for style transfer tasks. By focusing on the statistics of single instances, it helps preserve stylistic details and achieve more consistent visual outputs. Group Normalization Wu & He (2018) divides the channels of each layer into smaller groups and normalizes the features within each group. This method provides stable training benefits similar to BN but is less sensitive to mini-batch size, making it suitable for tasks with small mini-batch sizes. Mode Normalization Luo et al. (2019) adjusts the normalization process based on the mode of the feature distributions instead of their mean. This method aims to better handle skewed data distributions, resulting in improved training stability and model performance. Mixture Normalization Kalayeh & Shah (2019) addresses the limitations of BN in capturing the complex variations present in deep neural network activations. By leveraging Gaussian Mixture Models to assign samples to components and normalize based on multiple means and standard deviations, MN adapts to the diverse modes of variation inherent in the data distribution. RMSNorm Zhang & Sennrich (2019) extends Layer Normalization by utilizing the root mean square (RMS) of the activations within each layer. This method aims to stabilize training by normalizing activations based on their magnitudes, providing a robust normalization technique for deep neural networks. Unsupervised Batch Normalization Koçyigit et al. (2020) (UBN) leverages unlabeled examples to compute mini-batch statistics, addressing the challenge of bias on small datasets and offering regularization benefits from data manifold exploration. UBN demonstrates efficacy in tasks like monocular depth estimation, particularly beneficial where obtaining dense labeled data is challenging and costly.
While all these variants enhance the usability and stability of BN, our approach appears to be the first to extend BN by incorporating contexts, predefined groups of samples with shared characteristics, for normalization purposes.

### 2.2 INCORPORATING MULTIPLE MODES FOR EFFECTIVE NORMALIZATION

BN has been widely adopted in deep learning architectures to improve training stability and convergence. However, BN's assumption that the entire mini-batch should be normalized with the same mean and variance poses challenges, especially in the face of diverse data distributions. This assumption can lead to suboptimal performance, particularly on datasets with varying characteristics. Recent research has highlighted the limitations of this assumption, emphasizing the importance of accommodating multiple modes of variation within the data distribution. Approaches such as Mixture Normalization Kalayeh & Shah (2019), which employs Gaussian Mixture Models to capture multiple means and variances associated with different modes of variation, have been proposed to address this issue. Similarly, studies like Luo et al. Luo et al. (2019) have underscored the necessity of considering diverse data distributions and employing multiple mean and variance estimates for effective normalization. These insights emphasize the importance of moving beyond the simplistic assumptions of BN to better accommodate the complexities of real-world datasets.

## 3 METHOD

We begin by examining the formulations of BN with a single mode in Section 3.1, followed by an exploration of BN with multiple modes in Section 3.2. Finally, we present our method in Section 3.3.

### 3.1 BATCH NORMALIZATION WITH SINGLE MODE

Given an input mini-batch of height $H$ and width $W$ with $N$ samples and $C$ channels, represented as $x \in \mathbb{R}^{N \times C \times H \times W}$, BN normalizes each sample along the channel dimensions as follows:

$$\hat{x}_n = \gamma \left( \frac{x_n - \mu}{\sqrt{\sigma^2 + \epsilon}} \right) + \beta, \tag{1}$$

where $\mu$ and $\sigma^2$ represent the mean and variance respectively. Parameters $\gamma$ and $\beta$ are $C$-dimensional vectors aimed at learning an affine transformation along the channel dimensions, thereby preserving the representative capacity of each layer. while $\epsilon > 0$ serves as a small value to mitigate numerical instability.

The moving average of the mean $\bar{\mu}$ and variance $\bar{\sigma}^2$ are updated using a momentum rate $\alpha$ during training and used to normalize feature maps during inference:

$$\bar{\mu} = \alpha \bar{\mu} + (1 - \alpha)\mu \tag{2}$$

$$\bar{\sigma}^2 = \alpha \bar{\sigma}^2 + (1 - \alpha)\sigma^2 \tag{3}$$

When the samples within the mini-batch are drawn from the same distribution, the operation outlined in Equation 1 results in a distribution characterized by a mean of zero and a variance of one. This requirement for zero mean and unit variance acts to stabilize the activation distribution, thereby facilitating the training process. However, in scenarios where the samples stem from diverse distributions, a single mean and variance may prove insufficient, necessitating the adoption of strategies involving multiple modes (i.e., employing multiple means and variances) to achieve optimal results Kalayeh & Shah (2019); Luo et al. (2019).

### 3.2 BATCH NORMALIZATION WITH MULTIPLE MODES

The heterogeneous nature of complex datasets necessitates extending BN to multiple modes, enabling a more flexible and effective approach to normalization. A popular method that facilitates this is Mixture Normalization (MN) Kalayeh & Shah (2019). MN approaches BN from the perspective of Fisher kernels, derived from generative probability models. Instead of computing a single mean and variance across all samples within a mini-batch, MN employs a Gaussian Mixture Model (GMM) to assign each sample in the mini-batch to a component, then normalizes using multiple means and variances associated with different modes of variation in the underlying data distribution. Considering $K$ components, MN is implemented in two stages:

- Estimation of the mixture model's parameters $\theta = \{\lambda_k, \mu_k, \sigma_k^2 : k = 1, \ldots, K\}$ using the Expectation-Maximization (EM) algorithm Dempster et al. (1977).

- Normalization of each sample based on the estimated parameters and aggregation using posterior probabilities.

For a given input mini-batch $x \in \mathbb{R}^{N \times C \times H \times W}$, each sample $x_n$ is normalized along the channel dimensions as follows:

$$\hat{x}_n = \gamma \left( \sum_{k=1}^{K} \frac{p(k|x_n)}{\sqrt{\lambda_k}} \cdot \frac{x_n - \mu_k}{\sqrt{\sigma_k^2 + \epsilon}} \right) + \beta, \tag{4}$$

where

$$p(k|x_n) = \frac{\lambda_k p(x_n|k)}{\sum_{j=1}^{K} \lambda_j p(x_n|j)}$$

represents the probability that $x_n$ has been generated by the $k^{th}$ Gaussian component, with $p(x_n|k)$ and $\lambda_k$ denoting the density function of the Gaussian distribution and the mixture coefficient, respectively. The estimators for the mean $\mu_k$ and variance $\sigma_k^2$ are computed by weighting the contributions of $x_n$ ($\frac{p(k|x_n)}{\sum_j p(j|x_n)}$) with respect to the mini-batch when estimating the statistical measures of the $k$-th Gaussian component. Specifically, the $k$-th mean and variance are estimated from the mini-batch as follows:

$$\mu_k = \sum_n \frac{p(k|x_n)}{\sum_j p(j|x_n)} \cdot x_n \tag{5}$$

$$\sigma_k^2 = \sum_n \frac{p(k|x_n)}{\sum_j p(j|x_n)} \cdot (x_n - \mu_k)^2 \tag{6}$$

Multiple modes normalization methods extend Batch Normalization (BN) to heterogeneous complex datasets and often yield superior performance in supervised learning tasks. However, they are frequently computationally expensive due to tasks such as estimating different modes, such as the EM algorithm in Mixture Normalization (MN), and employing mixtures of experts Jordan & Jacobs (1994); Jacobs et al. (1991) in Mode Normalization.

To address the challenge of multiple modes and reduce computational costs compared to existing methods, we propose an approach that leverages prior knowledge to construct modes. This method significantly reduces costs while maintaining or even enhancing performance.

### 3.3 SUPERVISED BATCH NORMALIZATION

Our proposed method, SBN, introduces a novel approach to enhance neural network training efficiency. SBN operates by initially grouping samples into $K$ distinct contexts prior to training. Subsequently, during the training process, samples belonging to the same context $k$ within a given mini-batch are normalized using identical parameters $\mu_k$ and $\sigma_k^2$. By leveraging these predefined contexts, each comprising samples with similar characteristics, SBN effectively introduces multiple modes without incurring the computational overhead associated with estimating them during neural network training. This approach streamlines the normalization process and significantly reduces computational costs, thereby enhancing training efficiency and overall model performance.

#### 3.3.1 UNDERSTANDING CONTEXT: DEFINITION AND CONSTRUCTION METHODS

Context serves as the foundational element within SBN, representing groups of samples sharing similar characteristics. Our approach offers diverse methods for context construction:

- For domain adaptation tasks Zhang et al. (2021); Qi et al. (2020); Li et al. (2020), each domain is treated as a distinct context.

- In datasets featuring additional hierarchical structures, such as CIFAR-100 Krizhevsky et al. (2009a) or the Oxford-IIIT Pet dataset Parkhi et al. (2012), we designate each superclass as a separate context.

- For datasets lacking predefined contextual structures, we employ clustering algorithms like k-means Arthur & Vassilvitskii (2007) to partition samples into clusters, with each cluster forming an individual context.

This multifaceted approach ensures flexible and comprehensive context formation, vital for the effective implementation of SBN across various domains and datasets.

#### 3.3.2 TRAINING AND INFERENCE WITH SUPERVISED BATCH NORMALIZED NETWORKS

Consider $x \in \mathbb{R}^{N \times C \times H \times W}$ as a given input mini-batch and $K$ as the number of defined contexts. To normalize $x$, we first partition the samples in $x$ into $K$ groups based on their contexts, with each group $x^{(k)}$ containing samples that belong to context $k$. Each sample $x_n$ in $x^{(k)}$ is normalized using the same mean $\mu_k$ and variance $\sigma_k^2$ as given by Equation 4. Since each $x_n$ belongs to a single known context, $p(k|x_n) = 1$ if $x_n$ is in context $k$ and $p(k|x_n) = 0$ otherwise. Consequently, Equation 4 simplifies to:

$$\hat{x}_n = \gamma \left( \frac{1}{\sqrt{\lambda_k}} \cdot \frac{x_n - \mu_k}{\sqrt{\sigma_k^2 + \epsilon}} \right) + \beta, \tag{7}$$

where $\lambda_k$ represents the proportion of samples in the dataset belonging to context $k$. The mean and variance are then defined as follows:

$$\mu_k = \frac{1}{N_k} \cdot \sum_{n=1}^{N_k} x_n \tag{8}$$

---

**Algorithm 1:** Supervised Batch Normalization, training and inference phases

---

**Input** : $x = \{x_n\}_{n=1}^N$ : mini-batch of $N$ samples; $K$: number of contexts; $\{\gamma, \beta\}$: scale and shift learnable parameters; $\epsilon$: small value; $\alpha$: momentum; $\{\lambda_k\}_{k=1}^K$: proportion of samples in each context $k$; mode={Training, Inference}

**Output:** Normalized mini-batch $\{\hat{x}_n\}_{n=1}^N$

*// Training phase*

**if** *mode = Training* **then**

    **for** $k \leftarrow 1$ *to* $K$ **do**

        • Select the $N_k$ samples $x^{(k)}$ from $x$ that belong to context $k$

        • Compute the mean and variance:

$$\mu_k = \frac{1}{N_k} \cdot \sum_{n=1}^{N_k} x_n$$

$$\sigma_k^2 = \frac{1}{N_k} \cdot \sum_{n=1}^{N_k} (x_n - \mu_k)^2$$

        • Normalize each $x_n$ in $x^{(k)}$ :

$$\hat{x}_n = \gamma \left( \frac{1}{\sqrt{\lambda_k}} \cdot \frac{x_n - \mu_k}{\sqrt{\sigma_k^2 + \epsilon}} \right) + \beta$$

        • Compute the moving average of the mean and variance:

$$\bar{\mu}_k = \alpha \bar{\mu}_k + (1 - \alpha) \mu_k$$
$$\bar{\sigma}_k^2 = \alpha \bar{\sigma}_k^2 + (1 - \alpha) \sigma_k^2$$

    **end**

    *Replace the input mini-batch with the normalized mini-batch*

    **Return:** $\{\hat{x}_n\}_{n=1}^N$

**end**

*// Inference phase*

**if** *mode = Inference* **then**

    **if** *contexts are known* **then**

        **for** $k \leftarrow 1$ *to* $K$ **do**

            • Select all $x_n$ from $x$ that belong to context $k$

            • $\hat{x}_n = \gamma \left( \frac{1}{\sqrt{\lambda_k}} \cdot \frac{x_n - \bar{\mu}_k}{\sqrt{\bar{\sigma}_k^2 + \epsilon}} \right) + \beta$

        **end**

    **end**

    **if** *contexts are not known* **then**

        • Select all $x_n$ from $x$

        • $\hat{x}_n = \gamma \left( \sum_{k=1}^K \frac{p(k|x_n)}{\sqrt{\lambda_k}} \cdot \frac{x_n - \bar{\mu}_k}{\sqrt{\bar{\sigma}_k^2 + \epsilon}} \right) + \beta$

    **end**

    *Replace the input mini-batch with the normalized mini-batch*

    **Return:** $\{\hat{x}_n\}_{n=1}^N$

**end**

---

$$\sigma_k^2 = \frac{1}{N_k} \cdot \sum_{n=1}^{N_k} (x_n - \mu_k)^2 \tag{9}$$

where $N_k$ is the number of samples in the mini-batch that belong to context $k$.

The moving averages of the mean $\bar{\mu}$ and variance $\bar{\sigma}^2$ are updated with a momentum rate $\alpha$ during training. These updated values are then utilized to normalize feature maps during inference:

$$\bar{\mu}_k = \alpha\bar{\mu}_k + (1 - \alpha)\mu_k \tag{10}$$

$$\bar{\sigma}_k^2 = \alpha\bar{\sigma}_k^2 + (1 - \alpha)\sigma_k^2 \tag{11}$$

In the case where $K = 1$, it can be noted that SBN is equivalent to BN with a single mode.

During inference, for a given sample $x_n$, there are two possible normalization approaches. If the context of $x_n$ is known and identified as $k$, we normalize it using Equation 7 with the context-specific mean $\bar{\mu}_k$ and variance $\bar{\sigma}_k^2$. On the other hand, if the context of $x_n$ is unknown, we normalize it using Equation 4, which aggregates the normalization parameters across all $K$ contexts. This ensures that the sample is appropriately normalized regardless of whether its specific context is known.

The detailed steps for the training and inference phases of SBN are provided in Algorithm 1. This algorithm meticulously outlines the procedures for both phases, demonstrating how SBN normalizes mini-batches by leveraging context-specific grouping.

SBN extends BN to multiple modes without added cost by leveraging pre-defined contexts before training. Experiments on small datasets and classification tasks show improved convergence and performance compared to BN and other multi-mode normalization methods.

## 4    ANALYZING SBN IN A SIMPLIFIED SCENARIO

To demonstrate the principles behind SBN and its distinctions from BN, we conduct an experiment using a toy example. We train a simple 4-layer convolutional network with BN layers on the CIFAR-10 dataset Krizhevsky et al. (2009b). This dataset's simplicity allows for a deeper analysis, which would be challenging with a more complex task. For comparison, we create another model by replacing BN layers with SBN layers. To construct contexts for SBN, we use k-means clustering and vary the number of contexts across $K = \{2, 4, 6, 8\}$. Training is conducted on 50,000 data points with a fixed mini-batch size of 256. All models are trained for 100 epochs using the AdamW optimizer Loshchilov & Hutter (2017); Kingma & Ba (2014), with a weight decay parameter set to $10^{-4}$.

Table 1 demonstrates that SBN outperforms standard BN, indicating that incorporating multiple contexts is an effective method for normalizing intermediate features, even when the data is not heterogeneous.

Increasing the number of contexts $K$ does not affect performance, unlike other normalization

| model | 25 epochs | 50 epochs | 75 epochs | 100 epochs |
|-------|-----------|-----------|-----------|------------|
| BN    | 84,34     | 86,49     | 86,41     | 86,90      |
| SBN-2 | 85.56     | 87.62     | 87.70     | 87.70      |
| SBN-4 | 86.78     | 87.94     | 87.94     | 88.02      |
| SBN-6 | 86.79     | 88.00     | 88.48     | 88.56      |
| SBN-8 | **87.01** | **87.90** | **88.90** | **89.06**  |

Table 1: Test set accuracy rates (%) of batch normalization (BN) and supervised batch normalization (SBN) on the CIFAR-100 dataset. SBN-$k$ denotes SBN with $k$ contexts.

methods with multiple modes where increasing the number of modes can decrease performance. This is likely due to finite estimation, where estimates are computed from increasingly smaller batch partitions, a known issue in traditional BN.

## 5 EXPERIMENTS

We evaluate our methods in two experimental settings: (i) multi-task (heterogeneous dataset) and (ii) single task. To contrast with our proposed method SBN, we will utilize Batch Normalization (BN), Layer Normalization (LN), Instance Normalization (IN), Mixture Normalization (MN), and Mode Normalization (ModeN).

### 5.1 MULTI-TASK: UTILIZE EACH DOMAIN AS A CONTEXT

In this experiment, we demonstrate how SBN can significantly enhance domain adaptation by improving local representations. Domain adaptation involves leveraging knowledge from a related domain, where labeled data is abundant, to enhance model performance in a target domain with limited labeled data. We use two contexts ($K = 2$): the "source domain" and the "target domain". We apply normalization methods with AdaMatch, which combines unsupervised domain adaptation (UDA), semi-supervised learning (SSL), and semi-supervised domain adaptation (SSDA). In UDA, we use labeled data from the source domain and unlabeled data from the target domain to train a model that generalizes effectively to the target dataset. Notably, the source and target datasets have different distributions, with MNIST as the source dataset and SVHN as the target dataset, encompassing various factors of variation such as texture, viewpoint, and appearance.

A model, referred to as AdaMatch Paul (2019) (using BN layers), is trained from the ground up using wide residual networks Zagoruyko & Komodakis (2016) on pairs of datasets, serving as the baseline model. The training of this model involves utilizing the Adam optimizer Kingma & Ba (2014) with a cosine decay schedule, gradually reducing the initial learning rate initialized at 0.03. For comparison purposes, we substitute BN layers with LN, IN, MN, ModeN, and SBN. For MN and ModeN, determining the appropriate number of components and modes, respectively, involves conducting multiple tests. We retain the best results obtained with $K = 4$ for MN and $K = 3$ for ModeN.

Table 2 presents the test set performance rates (%) for various normalization methods in a

| MNIST (source domain) | | | |
|---|---|---|---|
| model | accuracy | precision | recall | f1-score |
| BN | 97.36 | 87.33 | 79.39 | 78.09 |
| LN | 96.23 | 88.26 | 76.20 | 81.70 |
| IN | **99.41** | **99.41** | **99.41** | **99.41** |
| MN | 98.90 | 98.45 | 98.89 | 98.93 |
| ModeN | 98.93 | 98.3 | 98.36 | 98.90 |
| SBN (ours) | 99.17 | 99.17 | 99.17 | 99.17 |
| SVHN (target domain) | | | |
| model | accuracy | precision | recall | f1-score |
| BN | 25.08 | 31.64 | 20.46 | 24.73 |
| LN | 24.10 | 28.67 | 22.67 | 23.67 |
| IN | 28.15 | 35.26 | 23.45 | 27.35 |
| MN | 32.14 | 50.12 | 37.14 | 39.26 |
| ModeN | 32.78 | 49.87 | 38.13 | 40.20 |
| SBN (ours) | **47.63** | **60.90** | **47.63** | 9.50 |

Table 2: Test set performance rates (%) for BN, LN, IN, MN, ModeN, and SBN on multi-task with heterogeneous dataset SVHN+MNIST for domain adaptation.

multi-task setting with the heterogeneous SVHN+MNIST dataset for domain adaptation. Notably, our proposed method, SBN, demonstrates significant improvements, particularly in the challenging SVHN target domain. Compared to BN, SBN achieves a remarkable gain in accuracy, with a **22.25**% increase. This highlights the efficacy of SBN in adapting to diverse datasets, even outperforming other normalization methods like MN and ModeN, which are based on multiple modes assumption. These results underscore the effectiveness of SBN in enhancing model performance across heterogeneous domains, making it a promising choice for domain adaptation tasks.

## 5.2 SINGLE TASK: UTILISE EACH SUPERCLASS AS A CONTEXT.

This experiment's main focus is on leveraging CIFAR-100 superclasses as contexts ($K = 20$) to predict the dataset's 100 classes, particularly with SBN. We utilize the base Vision Transformer model Dosovitskiy et al. (2020) obtained from Keras Salama (2021) as our baseline. To conduct comparisons, we modify this baseline by substituting different normalization layers. The training process includes early stopping based on validation performance, and image preprocessing involves normalization with respect to the dataset's mean and standard deviation. Additionally, data augmentation techniques such as horizontal flipping and random cropping are applied to enrich the dataset. To optimize model parameters and prevent overfitting, we employ the AdamW optimizer with a learning rate of $10^{-3}$ and a weight decay of $10^{-4}$ Loshchilov & Hutter (2017); Kingma & Ba (2014). Training is carried out for 100 epochs.

For Mixture Normalization (MN) and Mode Normalization (ModeN), determining the appropriate number of components and modes respectively involves conducting multiple tests. We save the best results (ref. Table 3) achieved with $K = 5$ for MN and $K = 3$ for ModeN.
Table 3 highlights the significant performance gains achieved by SBN compared to other normal-

| model | accuracy | precision | recall | f1-score |
|---|---|---|---|---|
| BN | 55.63 | 8.96 | 90.09 | 54.24 |
| LN | 54.05 | 11.82 | 85.05 | 53.82 |
| IN | 54.85 | 11.63 | 86.05 | 54.71 |
| MN | 53.2 | 11.20 | 87.10 | 54.23 |
| ModeN | 54.10 | 12.12 | 87.23 | 54.98 |
| SBN (ours) | **70.76** | **27.59** | **98.60** | **70.70** |

Table 3: Test set performance rates (%) for BN, LN, IN, MN, ModeN, and SBN on a single-task classification task using the CIFAR-100 dataset.

ization techniques (BN, LN, IN, MN, and ModeN). SBN shows a remarkable accuracy improvement of approximately **15.113%** over BN. It's worth noting that multiple modes normalization methods (MN, ModeN) do not perform well in this single-task scenario. However, by leveraging super-classes as contexts and normalizing accordingly, SBN outperforms all known ViT models trained from scratch on CIFAR-100. Figure 1 shows that SBN accelerates learning. These results indicate that SBN stabilizes data distributions, mitigates internal covariate shift, and significantly reduces training time for better outcomes.

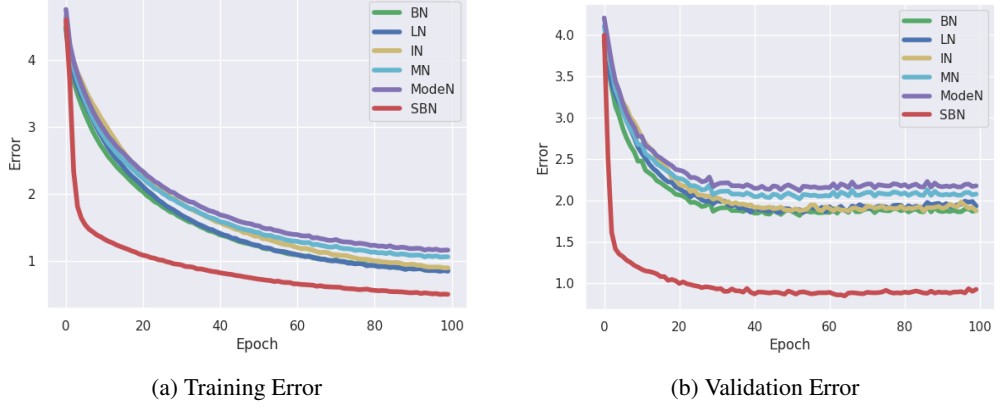

(a) Training Error                (b) Validation Error

Figure 1: Contrasting Training and Validation Error Curves in CIFAR-100 dataset

## 6 CONCLUSION

Our study introduces a groundbreaking normalization technique called Supervised Batch Normalization (SBN), which extends the capabilities of traditional Batch Normalization (BN) to effectively handle heterogeneous datasets characterized by diverse data distributions. Unlike BN, which normalizes each mini-batch using a single mean and variance, SBN addresses the challenge posed by varied data distributions within a mini-batch by normalizing based on grouped data with similar characteristics, referred to as contexts. We present three methods to accurately define these contexts.

Experimental results from both multi-task scenarios with heterogeneous datasets and single-task scenarios with homogeneous datasets demonstrate that SBN consistently outperforms BN and its variants, including methods based on multiple modes such as Mixture Normalization and Mode Normalization. SBN offers ease of implementation and versatility, serving as a powerful layer in neural networks to enhance performance and accelerate convergence.

Looking ahead, our future research will delve into exploring the robustness of SBN in multimodal systems, such as those involving text, image, audio, and other modalities, where contexts are well-defined and critical for effective normalization strategies.

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
