# OpenReview forum: "Supervised Batch Normalization"
_ICLR.cc/2025/Conference — Submitted to ICLR 2025_

### Official Review · Reviewer_JmWt · 2024-10-27

**Soundness:** 2
**Presentation:** 3
**Contribution:** 1
**Rating:** 3
**Confidence:** 4

**Summary:**

The paper “Supervised Batch Normalization” (SBN) introduces a novel extension to traditional Batch Normalization (BN). It aims to address BN’s limitations by grouping samples into predefined contexts based on shared characteristics (e.g., domains or clusters) and applying normalization specific to those contexts. This multi-mode normalization strategy improves performance in various tasks, such as classification on CIFAR-100 and domain adaptation between MNIST and SVHN. Notably, it reports 15.13% improvement on CIFAR-100 and a 22.25% increase in accuracy on SVHN compared to BN.

**Strengths:**

1. Originality: The idea of introducing predefined contexts within batch normalization offers a creative extension of BN.
2. Performance Gains: It achieves solid improvements on the datasets considered, particularly in multi-task domain adaptation.
3. Efficient Integration: The approach reduces computational overhead by leveraging predefined modes, avoiding costly estimations like those in Mixture Normalization.

**Weaknesses:**

1. Lack of Large-Scale Validation: The experiments on small datasets (MNIST, SVHN) cannot convincingly demonstrate the method’s efficacy. Larger datasets like ImageNet are crucial to validate whether SBN scales effectively in real-world applications.
2. Insufficient Baseline Comparisons: While SBN shows improvements over BN, comparisons with other advanced methods like Batch Renormalization are missing. Moreover, the results rely on limited normalization tasks, which weakens the generalizability of the findings.
3. Limited Exploration of Complex Contexts: The method assumes clear and well-separated contexts, but real-world scenarios often involve overlapping contexts (e.g., blurry domain boundaries in domain adaptation). The paper does not adequately address how SBN performs under these conditions. You should find or build a similar dataset to prove your contribution.
4. Experiments now I don't think will convince everyone in 2024 that your work is sufficiently productive and novel.

**Questions:**

1. Would the method show similar improvements on larger and more complex datasets like ImageNet?
2. How does the performance compare with Batch Renormalization or other alternatives in dynamic domain adaptation settings?
3. Can the method handle ambiguous or overlapping contexts effectively? How is normalization affected when contexts are not well-defined?

---

### Official Review · Reviewer_BASQ · 2024-10-28

**Soundness:** 2
**Presentation:** 3
**Contribution:** 2
**Rating:** 3
**Confidence:** 4

**Summary:**

This paper proposes a supervised batch normalization (SBN) method. This method splits various modes using the dataset's contexts.
Then, it builds a multi-mode BN to replace the traditional BN in the model.
The proposed method can be seamlessly integrated as layers in standard deep learning libraries.
And the authors have evaluated that the SBN outperforms BN and other widely used normalization techniques on various classification tasks.

**Strengths:**

1. Clear presentation
2. Significant improvement on baselines

**Weaknesses:**

1. Technical novelty is limited. The core algorithm of the SBN has been discussed in mixture BN. The innovation of this article lies only in using prior information to replace the original clustering center.

2. The experiments are unfair. For the supervised classification task, e.g., cifar100, SBN provides the superclass classification but the compared methods do not.

3. Some conclusions have not been fully substantiated. Like in Section 4, the authors claim "increasing the number of contexts K does not affect performance", however, this phenomenon is only present in one toy experiment, and no analytical evidence is provided.

4. The principle behind the effectiveness of this method has not been explained clearly. That is, why is SBN better than MN or ModeN with the context obtained from unsupervised clustering?

**Questions:**

Please see the weakness part.

---

### Official Review · Reviewer_FvVA · 2024-11-02

**Soundness:** 2
**Presentation:** 1
**Contribution:** 2
**Rating:** 3
**Confidence:** 4

**Summary:**

This paper studies the problem of batch normalization under diverse data distributions. A supervised batch normalization method is proposed to solve this problem, where the data modes are identified before training, and the mean and variances are used within the same context group. Experimental results on small datasets such as CIFAR, MNIST, and SVHN show the effectiveness of the proposed method.

**Strengths:**

- The issue of batch normalization under diverse data distributions is an important problem.
- The idea of context group for batch normalization is interesting. While the way to get the context group seems trival.

**Weaknesses:**

- During inference, when the contexts are not known, the mean value of statistics from all context groups is used, which is inconsistent with the training process. The inconsistency between training and testing should also be considered and might be more important than the gap among different context groups.

- In the experiments on CIFAR, k-means clustering is used to obtain the context group. While k-means cannot perform highly accurate classification, it is unclear if the quality of the context group obtained in this way is meaningful or not.

- The performance is only evaluated on small datasets such as CIFAR and MNIST. Experiments on large-scale datasets such as ImageNet should be conducted to verify the effectiveness of the proposed method.

- The domain adaptation ability of the proposed method is evaluated. While adapting the variance and mean of norms is widely used in domain adaptation methods, this method does not compare with those methods and is only evaluated on two toy datasets.

- For single-task evaluation, although there are many context-adaptive normalization methods, this method only compares with a few basic normalization schemes on small datasets. The effectiveness of the proposed method is unclear.

- The writing should be improved:
  - "then normalizes each sample based onin neural networks" — what does "onin neural networks" mean?

**Questions:**

- Verify the effectiveness on large-scale datasets and compare with strong baseline methods.
- Prove the effectiveness of context group generation.
- Compare the influence between training/testing inconsistency and context group inconsistency.

---

### Official Review · Reviewer_i6EV · 2024-11-03

**Soundness:** 2
**Presentation:** 2
**Contribution:** 2
**Rating:** 3
**Confidence:** 5

**Summary:**

The paper proposes Supervised Batch Normalization (SBN), a novel normalization technique aimed at enhancing the effectiveness of Batch Normalization (BN) for diverse data distributions. Traditional BN methods normalize each mini-batch using a single mean and variance, which may be suboptimal for heterogeneous data. SBN introduces “contexts”, which are predefined groups of similar samples, to normalize different subsets of data using multiple means and variances. The three methods for defining contexts in SBN are: by using predefined domains in domain adaptation tasks, grouping by superclasses, and clustering with algorithms like k-means for datasets without clear context divisions. Through experiments on both single-task (CIFAR-100) and multi-task (MNIST-SVHN) scenarios, SBN demonstrates significant improvements over BN and other normalization methods, achieving 15.13% and 22.25% accuracy improvement respectively.

**Strengths:**

* SBN effectively addresses limitations of traditional BN in handling heterogeneous data, offering a solution applicable across domains with varied data distributions.
* Experiments illustrate substantial performance gains, with SBN consistently outperforming other normalization methods on classification and domain adaptation tasks.
* The method integrates seamlessly with standard deep learning frameworks, making it accessible for practical implementation in various tasks.

**Weaknesses:**

* The process for constructing contexts is unclear and lacks detailed methodologies.
* The underlying principles behind SBN are insufficiently explained, making the approach less convincing.
* Using k-means clustering to define contexts in large-scale datasets demands substantial computation, potentially prolonging training time, and lacks experimental support in this paper.
* The novelty of SBN may be limited, as it could be perceived as a specific case of Mixture Normalization (MN) or Mode Normalization (ModeN).
* It is not clear how $\lambda$ and $p$ are calculated during inference.
* The paper contains several textual errors, such as “then normalizes each sample based onin neural networks, enhances generalization and expedites train- the statistics of its corresponding context” in Introduction section, issues with punctuation in Table 1, an incorrect f1-score for SBN in Table 2. Tables should not be inserted in the middle of a sentence.
* The cited literature lacks references from recent years, and SBN lacks comparison with recent methods.

**Questions:**

* The process for constructing contexts is unclear and lacks detailed methodologies.
* The underlying principles behind SBN are insufficiently explained, making the approach less convincing.
* The novelty of SBN may be limited, as it could be perceived as a specific case of Mixture Normalization (MN) or Mode Normalization (ModeN).

---

### Meta-Review · Area_Chair_W8JR · 2024-12-14

**Metareview:**

This paper proposed a supervised batch normalization, which actually has limited novelty. Similar supervision in batch normalization has been explored in [R1] Haifeng Xia, and Zhengming Ding. Cross-Domain Collaborative Normalization via Structural Knowledge. Thirty-Sixth AAAI Conference on Artificial Intelligence (AAAI), 2022.

The comparisons are not enough. The authors did not reply to the reviewers' comments.

**Additional Comments On Reviewer Discussion:**

The authors have not submitted any revision to reply the concerns from reviewers.

---

### Decision · Program_Chairs · 2025-01-22

Reject